# Postindustrial Landscapes Are Neglected Localities That May Play an Important Role in the Urban Ecology of Ticks and Tick-Borne Diseases—A Pilot Study

**DOI:** 10.3390/pathogens12050648

**Published:** 2023-04-27

**Authors:** Anna Dvořáková, Anita Klímová, Johana Alaverdyan, Jiří Černý

**Affiliations:** Centre for Infectious Animal Diseases, Faculty of Tropical AgriSciences, Czech University of Life Sciences Prague, Kamýcká 129, 165 00 Prague, Czech Republic

**Keywords:** ticks, postindustrial landscapes, vague terrain, urban landscapes, *Ixodes ricinus*, *Borrelia burgdorferi* sensu lato

## Abstract

Background: Numerous recent studies have shown that ticks and tick-borne pathogens pose a considerable threat in urban areas, such as parks, playgrounds, zoos, cemeteries, etc. Abandoned postindustrial localities, and other types of vague terrain, are other examples of urban wilderness areas that have been absolutely neglected in respect to ticks and tick-borne pathogens thus far, even though they provide ideal biotopes for ticks. Methods: The abundance of ticks and prevalence of *Borrelia burgdorferi* sensu lato spirochetes were compared between a city park and an adjacent abandoned construction waste disposal site in Prague, Czechia from June to October 2021. Results: The results showed that ticks and borrelia spirochetes are present at the city park as well as at the abandoned construction waste disposal site, although in lower numbers. Discussion: According to the best of our knowledge, this is the first report describing the presence of ticks and tick-borne pathogens in an urban postindustrial landscape. More detailed studies are needed to uncover the role of these localities in the ecology of ticks and ecoepidemiology of tick-borne diseases in urban areas.

## 1. Introduction

Ticks are important hematophagous parasites and vectors of numerous pathogens of veterinary and medical importance. Ticks and tick-borne pathogens are understood to be a problem connected with rural and natural areas, but numerous recent studies have shown that urban green areas may also provide suitable ecosystems for ticks, tick-borne pathogens, and their hosts/reservoirs [1,2,3]. These findings clearly demonstrate that urban localities, such as parks, playgrounds, zoos, and cemeteries, play an important role in the ecoepidemiology of ticks and tick-borne diseases [4,5,6,7]. Moreover, in these areas, ticks come in frequent contact with humans and their companion animals, which increases the risk of tick-borne pathogen transmission. 

On the other hand, risks connected with ticks and tick-borne diseases in urban ecosystems can be substantially decreased by employing proper management strategies within these localities [8]. Regular mowing and removal of organic debris (old grass, leaf litters), the creation of tick hostile ecosystems (light, dry, and sunny areas), control of tick-hosts, as well as area-wide use of acaricides or tick pathogens and parasitoids are just several examples of potential approaches for management of ticks and tick-associated risks in such areas [8]. Nevertheless, usage of these tick-management strategies is expensive, and therefore there are numerous localities within most cities that are undermanaged or not managed at all. 

Good examples of such localities are different types of vague terrain, which are places without any clear owners such as abandoned postindustrial landscapes and “spaces within spaces”—localities without any clearly defined public role, usually existing between other urban areas. These localities are typically transient in time, largely existing only for a few years or decades [9]. However, even a short period can provide enough time for the advanced succession and development of “urban jungle” ecosystems in these localities [10]. These localities can be also sources of many social problems, as they may attract homeless people and drug users, along with increased local criminality. On the other hand, they may also be used by local citizens to walk their pets, for community gardening, and many other beneficial purposes [9]. 

Despite the fact that these localities apparently offer an ideal ecosystem for ticks and tick-borne pathogens, there are no data about their role in the ecology of ticks and ecoepidemiology of tick-borne diseases, according to our best knowledge. The goals of this research were (i) to provide the first rigorous data about the presence of ticks and the prevalence of tick-borne pathogens in these localities and (ii) to attract the attention of tick-biologists, urban architects, local authorities, and other relevant entities to these localities, which may lead to the improvement of the current situation.

## 2. Materials and Methods

The study was conducted from June to October 2021, in two adjacent localities at the northwestern edge of Prague, Czechia: the city park Sedlecký sad (GPS: 50.127463, 14.390946), which is connected to large green areas within the Vltava River valley, outside the city of Prague, and a neighboring abandoned construction waste disposal site (50.130237, 14.388196) (Figure 1). Ticks were collected by flagging using approximately 1 × 1 m large flag made from white cotton. Flagging was performed on both localities several times during each month. The flagged localities within the boundaries of studied areas as well as our movement during flagging was completely random during each tick collection to prevent possible “overhunting” of ticks. Time spent at the locality and number of people who were flagging for ticks differed among the individual “tick hunts”. Therefore, the numbers of collected ticks were normalized according to the person-hours of collection. The presence of tick hosts at both localities was estimated using direct observation, camera traps, and observation of animal residential signs (Appendix A). Vegetation changes at both research sites during the tick season were photo-documented, and the most frequent species were determined using Google Lens (Appendix A). Borrelia were detected in the collected tick samples using PCR amplifying a fragment of the *flaB* gene, a marker of *Borrelia burgdoferi* sensu lato (s.l.) infection [11]. PCR results were visualized using agarose gel electrophoresis. Amplicons corresponding to the expected size were cut from the gel, extracted, and sent for sequencing to Biocev, Charles University in Prague. Obtained sequences were compared against sequences of known borrelia species in GenBank. Statistical analyses were performed in Statistica 12 (Tibco). Factorial ANOVA was used for comparison of average values of the collected ticks, depending on the location and month. An Χ2 test was used to compare prevalence of borrelia between the localities.

## 3. Results

### 3.1. The Abandoned Disposal of Nontoxic Construction Waste Material Shows Advanced Level of Biological Succession

The construction waste disposal site was used between 2001 and 2011 for the disposal of nontoxic construction waste material, such as rubble, grit, and sand. Since 2011, the locality was abandoned and left for “natural” succession. Therefore, ten years later, in 2021, the whole locality was covered with dense vegetation consisting of various species of grasses, small bushes, and young trees. Representatives of numerous animal species were observed at the abandoned construction waste disposal site including roe deer (*Capreolus capreolus*), red fox (*Vulpes vulpes*), pine marten (*Martes martes*), feral cat (*Felis catus*), ring-necked pheasant (*Phasianus colchicus*), and various songbirds, e.g., Eurasian magpie (*Pica pica*) and Eurasian jay (*Garrulus glandarius*). A similar composition of plant and animal species was observed in the unmanaged parts of the city park Suchdolský sad, but in the city park, there were also areas of twice-a-year mowed meadows, old fruit trees, small groves, and an old oak–beech forest. The animal richness was greater in the city park than at the construction waste disposal site, both in terms of the number of individuals as well as the number of species. In the city park, there was also higher activity of humans and companion animals (see Appendix A for photo documentation from both biotopes). 

### 3.2. Ticks Are Present at the Abandoned Disposal of Nontoxic Construction Waste Material Site, despite Being in Lower Abundance Compared to the Adjacent City Park

In total, 250 ticks were collected (92 from the abandoned construction waste disposal site and 158 from the city park—see Table 1 and Figure 2 for details). All collected ticks were morphologically determined as representatives of various developmental stages of the castor bean tick (Ixodes ricinus). Statistical analysis showed that a significantly higher number of ticks were present in the city park (*p* = 0.008622). This result can be explained by advanced, but not fully completed, succession at the abandoned construction waste disposal site and overall lower biological richness at that locality. Statistical analysis also showed a significant difference in number of ticks collected in each month (*p* = 0.000234).

### 3.3. Borrelia-Infected Ticks Are Presenet at the Abandoned Disposal of Nontoxic Construction Waste Material Site, despite Lower Prevalence Compared to the Adjacent City Park

In total, 20 tick samples were positive for *Borrelia burgdorferi* s.l. (4 ticks from the abandoned construction waste disposal site and 16 from the city park; see Appendix A for an example of a positive electrophoresis gel with adequate controls). The overall borrelia prevalence at the locality was 8%. A numerically greater borrelia prevalence was observed among the ticks collected from the city park (10.12%) than from the abandoned construction waste disposal site (4.34%), but this difference was not statistically significant (*p* = 0.10433). The entomological risk index (ERI) [12], calculated from the prevalence of borrelia positive ticks and overall abundance of questing ticks, was estimated to be 0.89 borrelia-infected ticks per hour for the city park and 0.21 borrelia-infected ticks per hour for the abandoned construction waste disposal site. The difference between ERI from both localities was not statistically significant (*p* = 0.130570).

Four species from the *Borrelia burgdorferi* s. l. complex were identified among the positive samples. The most prevalent species were *Borrelia burgdorferi* sensu stricto (10 samples) followed by *Borrelia afzelii* (6 samples), *Borrelia garinii* (2 samples), and *Borrelia lusitaniae* (2 samples) (see Table 2 for details). 

## 4. Discussion

Our results showed that ticks and borrelia spirochetes are present in the city park as well as in the adjacent abandoned construction waste disposal site, but tick numbers and overall prevalence of borrelia spirochetes were lower at the abandoned construction waste disposal site although not always significantly. We detected four species from the *Borrelia burgdorferi* s. l. complex, namely *Borrelia burgdorferi* sensu stricto, *Borrelia afzelii*, *Borrelia garinii*, and *Borrelia lusitaniae* (ordered according to prevalence at the study sites). This sequence is surprising, as in Czechia the highest prevalence is usually found for *B. garinii* and *B. afzelii*, followed by *B. burgdorferi* s.s. The observed discrepancy is most likely caused by the low number of positive samples. According to our best knowledge, this is the first research focused on tick abundance and tick-borne pathogen presence in urban postindustrial landscapes, which are otherwise neglected from a research perspective. Nevertheless, as shown in this study, they may pose an important but missing piece in the puzzle of our understanding of tick ecology and tick-borne disease ecoepidemiology in urban areas. The only similar research, performed on spoil banks resulting from brown coal mining in northern Bohemia, showed that ticks need about 20 years to colonize these localities and that this process can be accelerated by recultivation [13].

As mentioned earlier, many different people use postindustrial landscapes and other types of vague terrain [9], and therefore they can be infested by local ticks and subsequently infected by endemic tick-borne pathogens. Homeless people are probably the group with the highest risk, as they often camp in these localities and frequently suffer poor medical and social care. They usually also suffer from numerous chronic diseases (both infectious and noninfectious) including infection by borrelia spirochetes and other vector-borne diseases [14,15]. Homeless people are also at a higher risk of being infested by ticks and other ectoparasites, although lice and louse-borne diseases usually present higher problems than ticks and tick-borne diseases for homeless people [15,16]. As homeless people usually do not travel long distances, there is a high probability that the vast majority of the tick-borne diseases among them are caused by urban ticks, including those living in postindustrial vague terrain. A similar situation can be expected among other users of this type of landscape and among the animals spending at least part of their time there. 

Therefore, higher public awareness is needed to improve the current situation. Local citizens, the at-risk population, as well as local authorities and policy makers should be informed about tick-associated risk in urban areas, including postindustrial landscape and other types of vague terrain. This study should attract more tick-focused researchers to study postindustrial localities, which should provide the relevant data to local authorities to improve public health issues connected with vague terrains.

## Figures and Tables

**Figure 1 pathogens-12-00648-f001:**
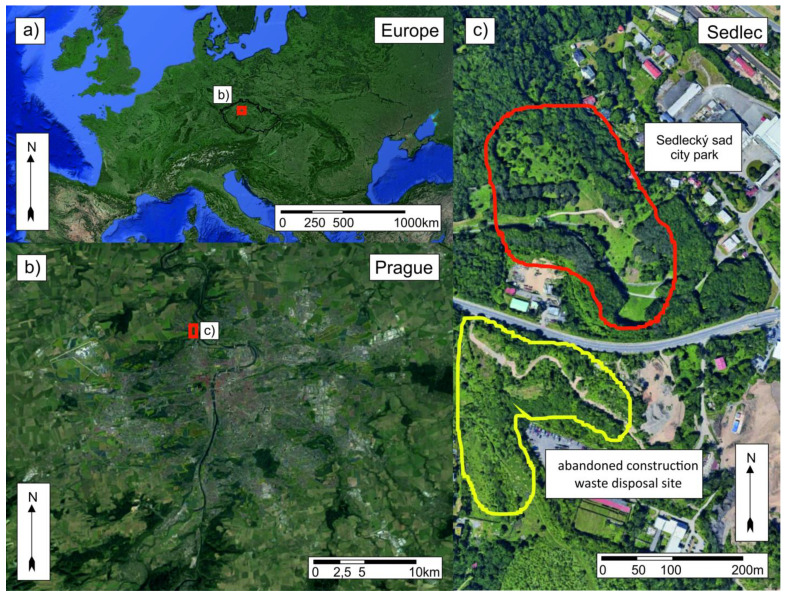
**Study site:** Location of the study site within the geographical context of (**a**) Europe, (**b**) Prague, and (**c**) the Sedlec neighborhood. The city park Sedlecký sad and the abandoned construction waste disposal site are adjacent areas separated only by Kamýcká street, which has a low traffic intensity, especially during the night, and therefore should pose almost no barrier to animal migration. The approximate flagged areas in the city park, Sedlecký sad, and the abandoned construction waste disposal site are marked with red and yellow lines, respectively.

**Figure 2 pathogens-12-00648-f002:**
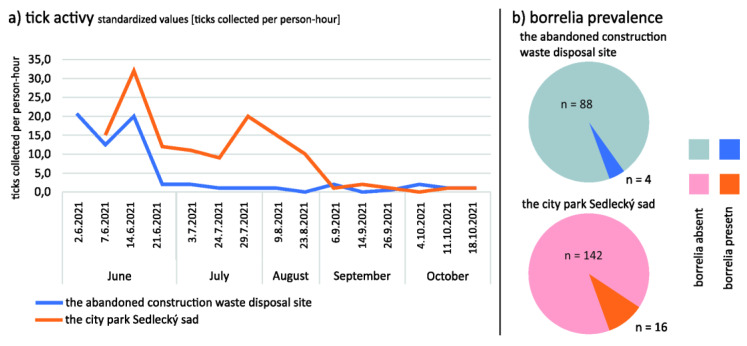
**Tick activity and prevalence of borrelia spirochetes:** (**a**) Numbers of ticks collected per person-hour in the city park, Sedlecký sad, (red line) and the abandoned construction waste disposal site (blue line) are indicated, during the tick season in 2021. (**b**) Prevalence of borrelia spirochetes within the collected tick specimens at both studied localities.

**Table 1 pathogens-12-00648-t001:** Ticks collected.

Locality	Month	Date	Flag. per. [h]	Pers.	Flag. [pers-h]	F	M	N	L	T	FS	MS	NS	LS	TS
The abandoned construction waste disposal site	June	02.06.2021	0.5	3	1.5	5	4	21	1	31	3.33	2.67	14	0.67	20.67
		07.06.2021	1	2	2	3	5	17	0	25	1.5	2.5	8.5	0	12.5
		14.06.2021	1	1	1	3	2	15	0	20	3	2	15	0	20
		29.06.2021	1	2	2	0	3	1	0	4	0	1.5	0.5	0	2
	July	03.07.2021	1	1	1	1	0	1	0	2	1	0	1	0	2
		24.07.2021	1	1	1	1	0	0	0	1	1	0	0	0	1
		29.07.2021	1	1	1	0	1	0	0	1	0	1	0	0	1
	August	09.08.2021	1	1	1	1	0	0	0	1	1	0	0	0	1
		23.08.2021	1	1	1	0	0	0	0	0	0	0	0	0	0
	September	06.09.2021	1	1	1	2	0	0	0	2	2	0	0	0	2
		14.09.2021	1	1	1	0	0	0	0	0	0	0	0	0	0
		26.09.2021	1	2	2	0	0	1	0	1	0	0	0.5	0	0.5
	October	04.10.2021	1	1	1	2	0	0	0	2	2	0	0	0	2
		11.10.2021	1	1	1	0	1	0	0	1	0	1	0	0	1
		18.10.2021	1	1	1	1	0	0	0	1	1	0	0	0	1
Subtotal for the abandoned construction waste disposal site					18.5	19	16	56	1	92	1.02	0.86	3.03	0.05	4.97
Sedlecký sad city park	June	02.06.2021	-	-	-	-	-	-	-	-	-	-	-	-	-
		07.06.2021	1	2	2	4	9	16	1	30	2	4.5	8	0.5	15
		14.06.2021	1	1	2	4	1	26	1	32	4	1	26	1	32
		29.06.2021	1	2	1	9	6	9	0	24	4.5	3	4.5	0	12
	July	03.07.2021	1	1	2	1	0	10	0	11	1	0	10	0	11
		24.07.2021	1	1	1	0	0	9	0	9	0	0	9	0	9
		29.07.2021	1	1	1	2	5	7	6	20	2	5	7	6	20
	August	09.08.2021	1	1	1	6	2	6	1	15	6	2	6	1	15
		23.08.2021	1	1	1	0	1	0	9	10	0	1	0	9	10
	September	06.09.2021	1	1	1	0	1	0	0	1	0	1	0	0	1
		14.09.2021	1	1	1	0	2	0	0	2	0	2	0	0	2
		26.09.2021	1	2	1	0	0	2	0	2	0	0	1	0	1
	October	04.10.2021	1	1	2	0	0	0	0	0	0	0	0	0	0
		11.10.2021	1	1	1	1	0	0	0	1	1	0	0	0	1
		18.10.2021	1	1	1	0	1	0	0	1	0	1	0	0	1
Subtotal for the city park					18	27	28	85	18	158	1.5	1.56	4.72	1	8.78

Flag. per. [h]—flagging period, Pers.—number of persons flagging, Flag. [pers-h]—flagging intensity [person-hours], F—females, M—males, N—nymphs, L—larvae, T—total ticks, FS—females-standardized [ticks per hour], MS—males-standardized [ticks per hour], NS—nymphs-standardized [ticks per hour], LS/larvae-standardized [ticks per hour], TS—total ticks-standardized [ticks per hour].

**Table 2 pathogens-12-00648-t002:** Detected borrelia species within the *Borrelia burgdorferi* sensu lato complex.

	*Borrelia burgdorferi* Sensu Stricto		*Borrelia afzelii*		*Borrelia lusitaniae*		*Borrelia garinii*		Subtotal Disposal Site	Subtotal City Park	Total
	Disposal Site	City Park	Disposal Site	City Park	Disposal Site	City Park	Disposal Site	City Park			
Females	1	2	1	1	0	0	0	0	2	3	5
Males	0	0	1	1	0	2	0	0	1	3	4
Nymphs	1	6	0	2	0	0	0	2	1	10	11
Larvae	0	0	0	0	0	0	0	0	0	0	0
Subtotal	2	8	2	4	0	2	0	2			
Total	10		6		2		2				20

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
