# Peer review of "Postindustrial Landscapes Are Neglected Localities That May Play an Important Role in the Urban Ecology of Ticks and Tick-Borne Diseases—A Pilot Study"

_pathogens, 2023, doi:10.3390/pathogens12050648_

Round 1

Reviewer 1 Report

The manuscript is well written and include important information of the ecology of Ixodes ricinus and the associated pathogen Borrelia burgdorferi in peri-urban areas. It would be fantastic to include data of and urban park of the city (not always but normally with limited medium and big mammals important in the ecology of I. ricinus) to compare the obtained data. Nevertheless, the data here showed are rigorous and remarks the importance of the areas that start to be naturalized near the urban areas, and the ricks of emergence of tick-borne diseases in the neighbouring population.

Thoughtful all the manuscript the authors refer to Borrelia burgdorferi sensu lato as Lyme borreliosis spirochetes (e.g. line 74, line 137) while the PCR used could detect all the group that includes some species have not been recognized as pathogens. It should be corrected.

Line 11: A comma should be added before etc.

The authors in the abstract indicate that the study was performed during the “2021 tick season”, but it is also done from June to October, while sprint is an important season, such as April and May (mainly for nymphs but also for other stages), and we also know that I. ricinus could be active all year. Please complete the period of time in the abstract.

Line 15: Use italic for Borrelia burgdorferi.

Line 66: The flagging method should be better explained, type of flag (material, area, time to time for the revision…). The area was flagging only in one direction? Is always sampled the same area? …

Line 74: The name of a gene should be in italic, and I think that flaB start also with lower caps.

Please correct the sentences in lines 105 to 109, some information is repeated and other incomplete.

In the title of 3.2., presenet mean “present”?

Line 120: The number of ticks is represented per person-hour. I think that the difference is not going to be high, but in my opinion it should be done by sampling area. With different persons and different times the ratio that the authors choose could not be the same. In the table we can see that the same point with different number of persons or different times takes till double of hours of sampling (Table), and it is clear that more area implies more collection and probably more time in the same area, but not in the same proportion. I could understand that the sampling was not done always in the same transect and it was done around all the area in the border, and the same area is not always sampled, it is that? It should be better explained in the methodology. If the sampling area is the same, the authors should shown the data according to it, easy to calculate when you have the area of the flag and the longitude of the transect that was checked.

Lines 140 and 141: decimals should be marked with “.” And not “,”

Line 149: This last sentence should be included as a discussion, not as a result

Table 2, title: Use italic for Borrelia burgdorferi.

In the table 2, please complete Borrelia burgdoferi sensu stricto, if not it seems that is the group and the ( 6 B. afzelii plus 2 B.lusitaniae plus 2 B. garinii)

Author Response

The manuscript is well written and include important information of the ecology of Ixodes ricinus and the associated pathogen Borrelia burgdorferi in peri-urban areas. It would be fantastic to include data of and urban park of the city (not always but normally with limited medium and big mammals important in the ecology of I. ricinus) to compare the obtained data.

Reply: Thank you very much for positive perception of our work. Regarding the study areas, similar objection was raised also by the Reviewer II. We totally agree that it would be great to have more research areas and to monitor them for a longer time. This was just a pilot study, which we also declare in the revised version od the manuscript title. In future we plan to do increase both number of observed areas as well as length of observation.

Nevertheless, the data here showed are rigorous and remarks the importance of the areas that start to be naturalized near the urban areas, and the ricks of emergence of tick-borne diseases in the neighbouring population.

Thoughtful all the manuscript the authors refer to Borrelia burgdorferi sensu lato as Lyme borreliosis spirochetes (e.g. line 74, line 137) while the PCR used could detect all the group that includes some species have not been recognized as pathogens. It should be corrected.

Reply: Corrected

Line 11: A comma should be added before etc.

Reply: Corrected

The authors in the abstract indicate that the study was performed during the “2021 tick season”, but it is also done from June to October, while sprint is an important season, such as April and May (mainly for nymphs but also for other stages), and we also know that I. ricinus could be active all year. Please complete the period of time in the abstract.

Reply: Corrected

Line 15: Use italic for Borrelia burgdorferi.

Reply: Corrected

Line 66: The flagging method should be better explained, type of flag (material, area, time to time for the revision…). The area was flagging only in one direction? Is always sampled the same area? …

Reply: Information about flagging were added to the manuscript: Flag dimensions were approximately 1 x 1 m large. Flag was made from white cotton. The flagged localities within the boundaries of studied areas as well as our movement during flagging was completely random during each tick collection to prevent possible “overhunting” of ticks.

Line 74: The name of a gene should be in italic, and I think that flaB start also with lower caps.

Reply: Corrected

Please correct the sentences in lines 105 to 109, some information is repeated and other incomplete.

Reply: Corrected, thank you for notifying us.

In the title of 3.2., presenet mean “present”?

Reply: Corrected, yes, that was a typo.

.

Line 120: The number of ticks is represented per person-hour. I think that the difference is not going to be high, but in my opinion it should be done by sampling area. With different persons and different times the ratio that the authors choose could not be the same. In the table we can see that the same point with different number of persons or different times takes till double of hours of sampling (Table), and it is clear that more area implies more collection and probably more time in the same area, but not in the same proportion. I could understand that the sampling was not done always in the same transect and it was done around all the area in the border, and the same area is not always sampled, it is that? It should be better explained in the methodology. If the sampling area is the same, the authors should shown the data according to it, easy to calculate when you have the area of the flag and the longitude of the transect that was checked.

Reply: We decided to normalize the number of collected ticks to person-hours as not only the sampled area but also the time spent on the locality and the number of flagging people were almost never the same. We understand that this is not the optimal set up but we used it to maximize the number of collected ticks for borrelia screening. Therefore we think that this way of normalizing the results is the best to straighten the objective differences between the individual “tick hunts”. These reasons are explained in the revised version of the manuscript.

Lines 140 and 141: decimals should be marked with “.” And not “,”

Reply: Corrected

Line 149: This last sentence should be included as a discussion, not as a result

Reply: Corrected

Table 2, title: Use italic for Borrelia burgdorferi.

Reply: Corrected

In the table 2, please complete Borrelia burgdoferi sensu stricto, if not it seems that is the group and the ( 6 B. afzelii plus 2 B.lusitaniae plus 2 B. garinii)

Reply: Corrected

Reviewer 2 Report

The purpose of this work is to spread awareness about the finding of Lyme disease ticks infected with borrelia spirochaetes in postindustrial landscapes and other types of vague terrains. I have the following comments.   Comments -    Why were samples collected only from two localities? The Sample size should have been collected from several post industrial landscapes or vague terrains.    Line 74-76 : Please add the gel pictures with positive and negative controls. .   Line 106 - It should be 'Richness' instead of 'richness'   Line 135 -It should be 'present' instead of 'presenet'

Author Response

The purpose of this work is to spread awareness about the finding of Lyme disease ticks infected with borrelia spirochaetes in postindustrial landscapes and other types of vague terrains. I have the following comments.  

Comments:

Why were samples collected only from two localities? The Sample size should have been collected from several post industrial landscapes or vague terrains.

Reply: Similar objection was raised also by the Reviewer I. We totally agree that it would be great to have more research areas and to monitor them for a longer period of time. This was just a pilot study, which we also declare in the revised version of the manuscript title. In future we plan to do increase both number of observed areas as well as length of observation.

Line 74-76 : Please add the gel pictures with positive and negative controls.

Reply: The requested photo was added to the manuscript as the Figure S2.

Line 106 - It should be 'Richness' instead of 'richness'

Reply: This sentence was removed during revision.

Line 135 -It should be 'present' instead of 'presenet'

Reply: Corrected